# XAIguiFormer:
# Explainable Artificial Intelligence Guided Transformer for Brain Disorder Identification

**Hanning Guo**[1,2], **Farah Abdellatif**[1,2], **Yu Fu**[3], **Jon. N. Shah**[1,4,5,6], **Abigail Morrison**[7,2],
**Jürgen Dammers**[1,8*]

[1] Institute of Neuroscience and Medicine (INM-4), Forschungszentrum Jülich, Germany

[2] Department of Computer Science 3 - Software Engineering, RWTH Aachen University, Germany

[3] School of Information Science and Engineering, Lanzhou University, China

[4] Department of Neurology, RWTH Aachen University, Germany

[5] JARA-BRAIN-Translational Medicine, Germany

[6] Institute of Neuroscience and Medicine (INM–11), JARA, Forschungszentrum Jülich, Germany

[7] Institute of Advanced Simulation (IAS-6), Forschungszentrum Jülich, Germany

[8] Department of Psychiatry, Psychotherapy and Psychosomatics, RWTH Aachen University, Germany

`{h.guo, j.dammers}@fz-juelich.de`

## Abstract

EEG-based connectomes offer a low-cost and portable method to identify brain disorders using deep learning. With the growing interest in model interpretability and transparency, explainable artificial intelligence (XAI) is widely applied to understand the decision of deep learning models. However, most research focuses solely on interpretability analysis based on the insights from XAI, overlooking XAI's potential to improve model performance. To bridge this gap, we propose a dynamical-system-inspired architecture, XAI guided transformer (XAIguiFormer), where XAI not only provides explanations but also contributes to enhancing the transformer by refining the originally coarse information in self-attention mechanism to capture more relevant dependency relationships. In order not to damage the connectome's topological structure, the connectome tokenizer treats the single-band graphs as atomic tokens to generate a sequence in the frequency domain. To address the limitations of conventional positional encoding in understanding the frequency and mitigating the individual differences, we integrate frequency and demographic information into tokens via a rotation matrix, resulting in a richly informative representation. Our experiment demonstrates that XAIguiFormer achieves superior performance over all baseline models. In addition, XAIguiFormer provides valuable interpretability through visualization of the frequency band importance. Our code is available at https://github.com/HanningGuo/XAIguiFormer.

## 1 Introduction

Scalp electroencephalography (EEG), an objective physiological method for detecting the human brain's neural activity, has been widely used in various applications, including disease diagnosis (Yang et al., 2024), seizure detection (Tang et al., 2022), action and motor imagery recognition (Venkatachalam et al., 2020), and emotional analysis (Yi et al., 2024). These non-invasive signals are collected by electrodes placed on the curved surface of the scalp, resulting in inherent non-Euclidean structures. The so-called connectome is a widely used metric to characterize the complex patterns and connectivity of the brain in this non-standard geometry (Pang et al., 2022). For instance, NeuroPath (Wei et al., 2024) is a graph transformer designed to model human connectomes and learn the relationship between neural pathways and brain functions. Additionally, EEG contains rich frequency domain information and is often analyzed over specific frequency bands. Consequently, multi-frequency band graphs are a powerful way to identify brain disorders using EEG signals.

---

[*]Corresponding author

In the past few years, transformer-based models have achieved great success in versatile applications, such as natural language processing (Vaswani et al., 2017), computer vision (Dosovitskiy et al., 2020), and brain signal processing (Yang et al., 2024). Across these diverse modalities, it is essential for transformers to design specific token generation methods based on the characteristics of the data. Existing approaches typically treat words, image patches, or time series segments as the basic units for embedding tokens. However, this segmentation strategy for tokenizing multi-band graphs disrupts their topological structure. Therefore, it is necessary to develop a novel tokenizer that divides multi-band graphs into atomic tokens while preserving sufficient graph structure.

Second, since transformers lack inherent mechanisms to capture the position of the input, explicitly injecting positional encoding is a common solution to perceive such latent information. However, is this necessary for multi-band connectomes deriving from EEG data? The primary purpose of positional encoding is to explicitly add information that transformers struggle to learn inherently, and this type of information is crucial for characterizing specific modalities. Accordingly, explicitly adding information to the input embeddings should take into account the intrinsic properties of the data. In the case of the multi-frequency band connectomes, each connectome inherently implies a specific frequency range and demographic information (i.e., age and gender) that also leads to significant anatomical, functional, and biochemical differences in the brain (Zaidi, 2010). However, conventional positional encoding methods fail to encode these intrinsic characteristics in EEG data. Therefore, another challenge is to inject implicit information, including frequency and individual differences due to demographics, into the transformer in a manner similar to positional encoding.

Third, the widespread application of transformers in medical diagnosis might be hindered by their black-box nature, as clinicians are concerned not only with the performance of transformers but also with the clinical rationale. To open the black box of deep learning models, explainable artificial intelligence (XAI) provides a promising solution for bringing transparency to complex and opaque deep learning models. Some interpretable models have emerged to offer various insights into a model's decision (Zhao et al., 2022; Li et al., 2021). However, almost all research in neuroimaging stops at the visualization purpose of explanation, neglecting to use this information for improving the performance of the transformer. On the other hand, transformers, especially large models, always benefit from transfer learning, knowledge distillation and convolutional operators. The common idea behind these techniques is to introduce additional knowledge from external datasets, teacher models or prior knowledge, respectively. Interestingly, XAI is also able to provide valuable insights to the transformer from the perspective of "self-refinement" without the demand for additional datasets and models (Stammer et al., 2023). In other words, XAI-based transformer improvement is perfectly suited for the limited amount of EEG data. Therefore, a key challenge is to take full advantage of XAI to improve the performance of the transformer.

To address the above-mentioned challenges, we propose XAI guided transformer (XAIguiFormer), a novel architecture for EEG-based brain disorder classification. XAIguiFormer takes multi-frequency band connectomes as input, constructed by two complementary methods (i.e. coherence (COH) and weighted phase lag index (wPLI), see Appendix A.3 for details) from preprocessed EEG signals. XAIguiFormer consists of three key components: connectome tokenizer, rotary frequency encoding with demographics (dRoFE), and XAI guided self-attention. The connectome tokenizer treats each single-band graph as an atomic token, forming a graph-wise frequency sequence. Compared to fragmenting the graph, our method can capture more comprehensive brain connectivity patterns in the frequency domain. In contrast to conventional positional encoding that adds position information to tokens, we explicitly integrate frequency and demographic information into tokens by the rotation matrix, thereby creating a more informative representation. To introduce valuable explanations to improve the transformer, XAIguiFormer employs XAI to refine the originally coarse query and key vectors within the self-attention mechanism, resulting in a more concentrated attention.

To the best of our knowledge, this is the first study employing XAI to enhance transformer performance in neuroimaging data, extending its benefits beyond mere interpretability analysis. The main contributions of this paper are as follows:

- we propose a novel approach to tokenize the connectome across frequency bands without destroying the topological structure of single-band graphs. It can be easily extended to other multi-graph scenarios, such as dynamic functional connectivity.
- we propose a fusion method to explicitly inject the frequency and demographic information into tokens, improving the model's understanding of frequency and mitigating the negative effects of individual differences.

- we propose to use XAI to directly enhance transformer performance rather than focusing only on analysing the visual interpretability. Inspired by the dynamical system, we propose an XAI guided self-attention mechanism along with an XAI guided loss function, directing the model's focus towards more relevant dependency relationships.

## 2  RELATED WORK

**Positional Encoding.**   Generally, there are two primary types of positional encoding for transformers. One is the absolute positional encoding (Vaswani et al., 2017), which injects the absolute position of tokens to the input embeddings through sinusoidal or learnable embeddings. The other is the relative positional encoding (Shaw et al., 2018), which encodes relative position information into the self-attention mechanism. Rotary Positional Embedding (RoPE) (Su et al., 2024) is a special relative position embedding that uses the absolute position to represent the relative position via a rotation matrix. In addition to these general-purpose methods, several task-specific positional encodings have been developed to exploit the unique properties of specific types of data. For instance, Informer (Zhou et al., 2021) considers the intrinsic characteristics of time series, such as week, month, year and holidays, and proposes hierarchical (local-global) time stamps as positional encoding to forecast long sequence time series. Brain-JEPA (Dong et al., 2024) also considers the inherent functional relationships among brain regions, thereby developing brain gradient positioning to inject temporal, spatial and functional information into tokens. In the EEG field, MMM (Yi et al., 2024) introduces multi-dimensional position encoding to capture intrinsic geometric layout of electrodes. To preserve the spatiotemporal features of EEG data, BIOT (Yang et al., 2024) adds channel embedding and relative position embedding to each token. However, those techniques fail to integrate frequency bands and individual differences into tokens, causing missing essential information in multi-band connectomes.

**Explainable Artificial Intelligence.**   The increasing demand for model transparency and trustworthiness in recent years has driven the development of various explanation methods for black-box deep learning models. A prominent category is the saliency method, which identifies the most important input features contributing to the model's decision, such as LRP (Bach et al., 2015) and SHAP (Lundberg & Lee, 2017). In contrast, some inherently interpretable deep learning models have been proposed to provide explanations during training, such as CRATE (Yu et al., 2024) and BrainGNN (Li et al., 2021). Although these visualizations of explanations can help researchers identify flawed model reasoning, they often lack the ability to directly guide improvements in the model's internal parameters or architecture during training. To this end, Weber et al. (2023) conducted a comprehensive overview of XAI techniques for enhancing various properties of deep learning models, demonstrating how explanations can be leveraged to improve components in deep learning models, including data, feature representations, loss functions, gradients, and trained models. However, few studies have fully utilized XAI to improve transformers.

## 3  PRELIMINARY KNOWLEDGE

**Rotary Positional Embedding.**   The RoPE is a novel position embedding method originally designed for natural language processing to encode the relative positions among tokens in a sequence, which is widely applied in large language models. The core idea behind RoPE is to rotate the query $q$ and the key $k$ by certain angles to characterize the position $m$. In the complex domain, the rotary matrix is defined as

$$R_{\Theta,m}^d = e^{im\Theta} \tag{1}$$

where $d$ is the embedding dimension, $i$ is the imaginary number, and $\Theta$ is a pre-defined parameter given by

$$\Theta = \{\theta_t = 10000^{-2t/d} \mid t \in \{0, 1, \cdots, d/2 - 1\}\} \tag{2}$$

The rotary matrix is then applied to $q$ and $k$ for injecting the position information by

$$f_{\{q,k\}}(\boldsymbol{x}_m; W_{\{q,k\}}) = R_{\Theta,m}^d W_{\{q,k\}} \boldsymbol{x}_m \tag{3}$$

where $\boldsymbol{x}_m$ is the $m$-th token in a sequence and $W_{\{q,k\}}$ represents the weights specific to the $q$ and $k$. Finally, the attention value with RoPE is calculated by

$$A_{m,n} = Re < f_q(\boldsymbol{x}_m; W_q), f_k(\boldsymbol{x}_n; W_k) >$$

where $Re < \cdot >$ denotes the real part of the complex number. In this formulation, RoPE naturally incorporates relative position information through the inner product of the rotation matrix, which exhibits remarkable performance.

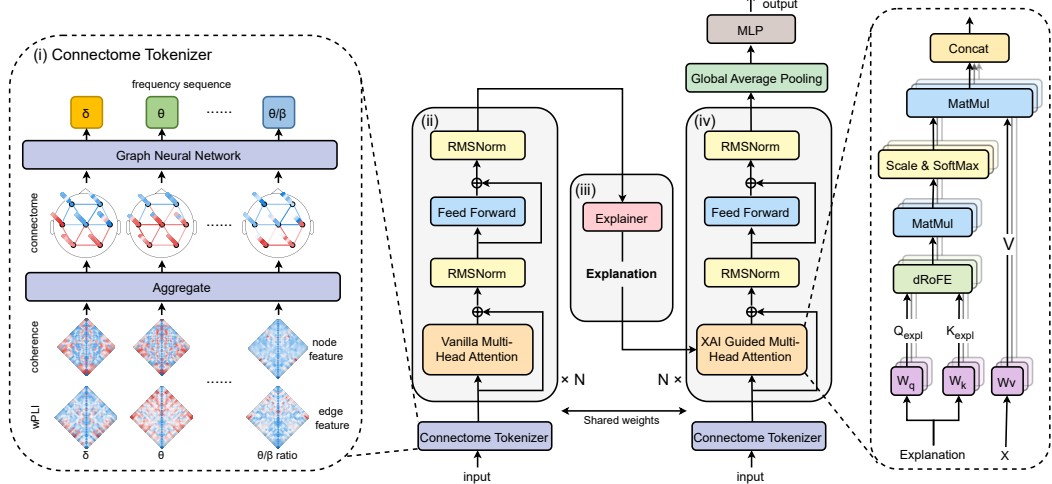

Figure 1: The architecture of XAIguiFormer. XAIguiFormer forward process can be described as follows: (i) construct multi-frequency band connectomes and generate a sequence in the frequency domain by connectome tokenizer, (ii) forward pass the frequency sequence by vanilla transformer, (iii) obtain refined features by explaining the vanilla transformer, (iv) feedforward refined features and the frequency sequence through XAI guided transformer. Subsequently, the MLP is employed as the classification head to predict the label of the brain disorder.

**Understanding Self-attention Mechanism via Dynamical System Perspective.** Huang et al. (2023) explain the self-attention mechanism from the mathematical perspective of dynamical system. In such a framework, the transformer with residual block can be written as

$$\hat{\boldsymbol{x}}_{t+1} = \boldsymbol{x}_t + \underbrace{f_v(\boldsymbol{x}_t; W_v)}_{Feature\ map} \otimes \underbrace{\phi(f_q(\boldsymbol{x}_t; W_q), f_k(\boldsymbol{x}_t; W_k))}_{Attention\ value} \tag{4}$$

where $\boldsymbol{x}_t$ is the input sequence of the linear layer $f(\cdot; W_{\{q,k,v\}})$ with the learnable parameters $W_{\{q,k,v\}}$ in the $t$-th block, $\otimes$ denotes the Hadamard product and $\phi$ is the self-attention module. Further, equation 4 can be reformulated as an ordinary differential equation (ODE)

$$f_v(\boldsymbol{x}_t; W_v) = \frac{\boldsymbol{x}_{t+1} - \boldsymbol{x}_t}{\Delta t} \tag{5}$$

Here, $\Delta t$ can be interpreted as the attention value generated by the self-attention module $\phi(f_q, f_k)$, which serves as an adaptive step size in ODE. Assuming $f_{\{q,k,v\}}$ are all invertible, the attention value function $\phi$ can be transformed to

$$\phi(f_q(\boldsymbol{x}_t; W_q), f_k(\boldsymbol{x}_t; W_k)) = \phi(f_q \circ f_v^{-1}(v), f_k \circ f_v^{-1}(v)) \stackrel{\Delta}{=} \tilde{\phi}(v) = \tilde{\phi}(\underbrace{f_v(\boldsymbol{x}_t; W_v)}_{Coarse\ Info}) \tag{6}$$

Here, $f_v(\boldsymbol{x}_t; W_v) = \frac{1}{\Delta t}(\boldsymbol{x}_{t+1} - \boldsymbol{x}_t)$ is a kind of coarse stiffness information in feature trajectory at $\boldsymbol{x}_t$. Taking into account all these equations, it can be concluded that the attention value function $\phi$ takes coarse information $f_v(\boldsymbol{x}_t; W_v)$ as input and refines it to obtain a finer estimation of information, further generating adaptive attention values. This implies the importance of establishing better input for self-attention and the resulting attention values to capture refined information in the transformer.

## 4 METHODOLOGY

In this study, we introduce the XAI guided transformer, a novel architecture that not only interprets the model's decisions but also utilizes explanations to enhance the performance of the transformer in an end-to-end fashion. Instead of only considering the position in the sequence, our model integrates the frequency and demographic information into tokens by a specially designed rotary matrix. As illustrated in Figure 1, the model architecture consists of 4 steps: (i) generate a token for each frequency band connectome using the graph neural network, (ii) pass the frequency sequence through

the vanilla transformer encoder while inject the frequency and demographic information to tokens by dRoFE, (iii) explain the vanilla transformer to obtain refined features, (iv) run the feedforward path again by feeding refined query and key to XAI guided transformer. Finally, the XAI guided loss function is employed to train the entire model.

## 4.1 PROBLEM FORMULATION

Let the multi-graph $\mathcal{G} = \{\mathcal{G}_\delta, \mathcal{G}_\theta, \ldots, \mathcal{G}_{\theta/\beta}\}$ represent the brain connectomes across various frequency bands $Freq = \{\delta, \theta, low\ \alpha, high\ \alpha, low\ \beta, mid\ \beta, high\ \beta, low\ \gamma, \theta/\beta\ ratio\}$. For each frequency band connetome $\mathcal{G}_f = (\mathcal{V}, \mathcal{E}_f)$, EEG channels are defined as graph nodes $\mathcal{V} = \{v_1, v_2, \ldots, v_c\}$, where $c$ is the number of EEG channels and $f \in Freq$. The node features at frequency band $f$ are denoted by $\boldsymbol{H}^f_{COH} = [\boldsymbol{h}^f_{v_1}, \boldsymbol{h}^f_{v_2}, \ldots, \boldsymbol{h}^f_{v_c}]^T$, where $\boldsymbol{h}^f_{v_i}$ is the coherence-based connectivity patterns of node $v_i$ at frequency band $f$ (see Appendix A.3 for details). The edge features at frequency band $f$ are indicated by $\boldsymbol{E}^f_{wPLI} = [e^f_{i,j}]$, where $e^f_{i,j}$ is the wPLI-based connectivity between nodes $v_i$ and $v_j$ at frequency band $f$. The adjacency matrix of the single-band graph $\mathcal{G}_f$ is represented as $\boldsymbol{A}_f \in \mathbb{R}^{c \times c}$. In addition, let $Demog = [age,\ gender]$ be the demographic information. The main objective of this study is to establish the mapping from $(\mathcal{G}, Demog)$ to brain disorder diagnosis $Y$ by learning a function $F : (\mathcal{G}, Demog) \rightarrow Y$.

## 4.2 CONNECTOME TOKENIZER

A fundamental operation for transformers is the design of tokenizers tailored to specific data modalities. For images, ViT splits the image into a sequence of flattened patches (Dosovitskiy et al., 2020). Although the multi-frequency band connectome $\mathcal{G}$ can be regarded as a multi-channel "image" due to its grid-like structure, partitioning the connectome into patches risks destroying its connectivity and results in the loss of topological information. Inspired by the tokenization method of DETR (Carion et al., 2020), we adopt a strategy where the entire single-band connectome $\mathcal{G}_f$ serves as the basic unit to generate tokens. To this end, the GINEConv operator (Hu et al., 2020) is employed to update node representations at $l$-th layer by

$$\boldsymbol{h}^{(l+1)}_{v_i,f} = \mathrm{MLP}((1 + \epsilon) \cdot \boldsymbol{h}^{(l)}_{v_i,f} + \sum_{v_j \in \mathbb{N}(v_i)} \mathrm{ReLU}(\boldsymbol{h}^{(l)}_{v_j,f} + e^f_{j,i})) \qquad (7)$$

where $\mathbb{N}(v_i)$ represents the set of nodes adjacent to $v_i$ and $\epsilon$ is a trainable parameter. Notably, leveraging the GNN also introduces a beneficial inductive bias to our model, particularly on relatively small datasets. Subsequently, a single-band token $\boldsymbol{x}_f$ is obtained by averaging the node embeddings at the last layer of GNN using

$$\boldsymbol{x}_f = \mathrm{MEAN}(\{\boldsymbol{h}^f_{v_i} \mid v_i \in \mathcal{V}\}) \qquad (8)$$

Finally, those token embeddings $\boldsymbol{x}_f$ are flattened into a frequency sequence $\boldsymbol{X} = [\boldsymbol{x}_\delta, \boldsymbol{x}_\theta, \ldots, \boldsymbol{x}_{\theta/\beta}]^T \in \mathbb{R}^{|Freq| \times d}$, which are then fed into the XAI-guided transformer encoder.

## 4.3 ROTARY FREQUENCY ENCODING WITH DEMOGRAPHICS

Conventional RoPE only considers intrinsic positional properties of the sequential data in text and image. However, positional information is less valuable in the frequency sequence $\boldsymbol{X}$ because its order can be arbitrary, and treating $\boldsymbol{X}$ as a simple sequence risks losing the intrinsic frequency information. Hence, we encode the frequency information by rotating each frequency band embedding with angles determined by frequency bounds

$$R^d_{f,2t} = e^{if_l\theta_t},\ \ R^d_{f,2t+1} = e^{if_u\theta_t} \qquad (9)$$

where $f_l$, $f_u$ is the lower and upper bound of the frequency band $f$, and $i$ is the imaginary number. To achieve smoother frequency encoding in the short sequence $\boldsymbol{X}$ compared to the exponentially scaled one in the language sequence, the $\theta$ at the $t$-th dimension of tokens is defined as

$$\theta_t = 4\pi t/d,\ t \in \{0, 1, \ldots, d/4 - 1\} \qquad (10)$$

Moreover, age and gender are known to influence differences in individual connectomes (Mijalkov et al., 2023). To alleviate the individual differences, we propose to integrate **d**emographics into

rotary frequency encoding (dRoFE) by encoding demographic information as the magnitude of the vector. Taking together, the rotation matrix of dRoFE is defined as

$$R_{f,Demog,2t}^{d} = age \cdot e^{if_l\theta_t} + gender, \quad R_{f,Demog,2t+1}^{d} = age \cdot e^{if_u\theta_t} + gender \qquad (11)$$

Finally, dRoFE is applied on the query $q$ and key $k$ to explicitly inject frequency and demographic information by

$$dRoFE_{\{q,k\}}(\boldsymbol{x}_f, f, Demog) = R_{f,Demog}^{d} W_{\{q,k\}} \boldsymbol{x}_f \qquad (12)$$

In practical implementation, dRoFE introduces the multiplication of Euler's formula to $q$ and $k$, which can produce the same attention value $\boldsymbol{q}\boldsymbol{k}^T = \boldsymbol{Re}[q_i k_i^*]$ but reduces computational burden (Heo et al., 2024). Here, $\boldsymbol{Re}[\cdot]$ denotes the real part of the complex number and $q_i$, $k_i$ refer to the complex vectors.

### 4.4 XAI GUIDED TRANSFORMER

**XAI Guided Self-Attention.**    According to the insights from the section 3 preliminary knowledge, the ability to properly generate refined input and attention value is essential for the performance of the self-attention mechanism. To enhance the transformer's ability to capture refined information, we propose a theoretic-inspired self-attention mechanism, named XAI guided self-attention. The value of XAI lies in its capability to identify important features and filter out irrelevant information for self-attention input. Consequently, the XAI-generated importance maps for $\boldsymbol{Q}$ and $\boldsymbol{K}$ are more refined than the original ones. Further, the attention value is calculated based on these underlying important features identified by XAI, effectively removing noise to generate a more precise attention distribution that concentrates on important dependency relationships by

$$Attn(\boldsymbol{Q}_{expl}, \boldsymbol{K}_{expl}, \boldsymbol{V}) = softmax\left(\frac{\boldsymbol{Q}_{expl}\boldsymbol{K}_{expl}^T}{\sqrt{d}}\right)\boldsymbol{V} \qquad (13)$$

where $\boldsymbol{Q}_{expl}$ and $\boldsymbol{K}_{expl}$ are the feature importance scores generated by DeepLift (Shrikumar et al., 2017). To access those valuable explanations, we first process the frequency sequence $\boldsymbol{X}$ via the vanilla transformer encoder. We then employ DeepLift to pinpoint which parts are most influential for the vanilla transformer. In general, the explanation algorithm can be realized with any local explanation method from XAI. By making use of XAI, we can improve originally coarse information to capture more focused attention value, ultimately enhancing the representative ability of the transformer.

Finally, the XAIguiFormer model comprises a stack of several encoder blocks. As illustrated in Figure 1, each XAIguiFormer block consists of an XAI guided multi-head self-attention module, followed by a 2-layer MLP with GeGLU activation function. A residual connection is applied after XAI guided multi-head attention and the feed forward network, followed by an RMSNorm layer for normalization.

**XAI Guided Loss Function.**    To boost the performance of model under the supervision of XAI, we propose a joint loss function to optimize the parameters

$$\mathcal{L} = \underbrace{(1-\alpha)\,\mathcal{L}_{CE}\,(\,\hat{\boldsymbol{y}}_{crs}, \boldsymbol{y})}_{Coarse\ Prediction} + \underbrace{\alpha\mathcal{L}_{CE}\,(\hat{\boldsymbol{y}}_{expl}, \boldsymbol{y})}_{Refined\ Prediction} \qquad (14)$$

where $\mathcal{L}_{CE}$ refers to cross-entropy loss, $\hat{\boldsymbol{y}}_{crs}$ and $\hat{\boldsymbol{y}}_{expl}$ denote the labels predicted by vanilla and XAI guided self-attention mechanisms, respectively. The parameter $\alpha$ controls the magnitude of XAI guidance. In our experiments, XAIguiFormer is not very sensitive to $\alpha$ as long as it is larger than 0.3.

## 5 EXPERIMENTS

### 5.1 DATASETS

We evaluated the performance of our XAIguiFormer model on two publicly available datasets:

• The TUH Abnormal EEG Corpus (TUAB) is one of the largest open-source EEG datasets, which collects data from 1385 healthy controls (normal) and 998 subjects having a brain disorder (abnormal), with a total of 2993 sessions (Obeid & Picone, 2016). All electrophysiology recordings contain 19 common EEG channels and 2 reference channels across subjects. We employ the XAIguiFormer to perform binary classification for identifying brain disorders on this dataset.

Table 1: Comparison of model performance with baseline methods on the TUAB dataset.

| Methods | Model Size | FLOPs | TUAB | | |
|---|---|---|---|---|---|
| | | | BAC | AUC-PR | AUROC |
| FFCL (2022b) | 2.4M | 0.83G | 0.7848 ± 0.0038 | 0.8448 ± 0.0065 | 0.8569 ± 0.0051 |
| SPaRCNet (2023) | 0.79M | 0.26G | 0.7896 ± 0.0018 | 0.8414 ± 0.0018 | 0.8676 ± 0.0012 |
| BIOT (2024) | 3.2M | 1.9G | 0.7959 ± 0.0057 | 0.8792 ± 0.0023 | 0.8815 ± 0.0043 |
| S3T (2021) | 3.5M | 0.22G | 0.7966 ± 0.0023 | 0.8521 ± 0.0026 | 0.8707 ± 0.0019 |
| LaBraM-Base (2024) | 5.8M | 2.7G | 0.8140 ± 0.0019 | 0.8965 ± 0.0016 | **0.9022 ± 0.0009** |
| Corr-DCRNN (2022) | 0.69M | 0.21G | 0.7672 ± 0.0050 | 0.8586 ± 0.0044 | 0.8564 ± 0.0040 |
| LGGNet (2024) | 1.4M | 0.64G | 0.7711 ± 0.0030 | 0.8594 ± 0.0010 | 0.8574 ± 0.0011 |
| XAIguiFormer (Ours) | 3.5M | 1.6G | **0.8205 ± 0.0027** | **0.8965 ± 0.0079** | 0.9000 ± 0.0046 |

Table 2: Comparison of model performance with baseline methods on the TDBRAIN dataset.

| Methods | Model Size | FLOPs | TDBRAIN | | |
|---|---|---|---|---|---|
| | | | BAC | AUC-PR | AUROC |
| FFCL (2022b) | 2.4M | 0.83G | 0.5394 ± 0.0400 | 0.4258 ± 0.0278 | 0.5583 ± 0.0490 |
| SPaRCNet (2023) | 0.79M | 0.26G | 0.4786 ± 0.0153 | 0.3448 ± 0.0194 | 0.4580 ± 0.0224 |
| BIOT (2024) | 3.2M | 1.9G | 0.6320 ± 0.0082 | **0.6007 ± 0.0205** | 0.7175 ± 0.0176 |
| S3T (2021) | 3.5M | 0.22G | 0.4937 ± 0.0214 | 0.3500 ± 0.0174 | 0.4739 ± 0.0195 |
| LaBraM-Base (2024) | 5.8M | 2.7G | 0.6456 ± 0.0089 | 0.5438 ± 0.0058 | 0.7147 ± 0.0145 |
| Corr-DCRNN (2022) | 0.69M | 0.21G | 0.6045 ± 0.0058 | 0.4951 ± 0.0131 | 0.6586 ± 0.0158 |
| LGGNet (2024) | 1.4M | 0.64G | 0.6152 ± 0.0098 | 0.5058 ± 0.0161 | 0.6929 ± 0.0174 |
| XAIguiFormer (Ours) | 3.5M | 1.6G | **0.6635 ± 0.0080** | 0.5961 ± 0.0136 | **0.7814 ± 0.0156** |

• The Two Decades-Brainclinics Research Archive for Insights in Neurophysiology (TDBRAIN) is a clinical lifespan EEG biobank comprising diverse patients with major depressive disorder, attention deficit hyperactivity disorder and obsessive-compulsive disorder (Van Dijk et al., 2022). We utilized eyes-closed condition with 26 channels of EEG data to carry out multi-class classification to distinguish brain disorders.

For further details on the demographic distribution and preprocessing steps of those datasets, as well as the construction method of XAIguiFormer's input connectomes, please refer to Appendix A.

Additionally, details on the baseline models and evaluation metrics can be found in Appendix B.

## 5.2 IMPLEMENTATION DETAILS

Our experiments are implemented using Python 3.10.12, PyTorch 2.0.1+CUDA11.8 and PyTorch-Geometric 2.3.1 on a single NVIDIA A100 GPU. The connectome input dimensions are 19 for TUAB and 26 for TDBRAIN, both of which are mapped into 128 as hidden dimensions. The model consists of a 4-layer GNN and a 12-layer XAIguiFormer with 4 attention heads, training by AdamW optimizer with $\beta = (0.9, 0.99)$ and $\epsilon = $ 1e-8. For the TUAB dataset, the learning rate is set to 4e-5 with a weight decay of 1e-4, while for the TDBRAIN dataset, the learning rate is set to 5e-5 with a weight decay of 1e-5. The loss weight for XAI-guided training is set to 0.7. Due to the different sizes of the TUAB and TDBRAIN datasets, the batch sizes are set to 512 and 64, respectively. More details about hyperparameters can be found in Appendix G.

## 5.3 COMPARISON WITH BASELINE METHODS

Table 1 and 2 present the performance comparison between XAIguiFormer and baseline models on the TUAB and TDBRAIN datasets, where we report subject-wise performance by averaging the predictions of all 30-second samples belonging to the same subject. The results demonstrate that XAIguiFormer outperforms all baselines on various evaluation metrics for both datasets, except for the AUC-PR in the TDBRAIN dataset. In particular, on the relatively small TDBRAIN dataset, the XAIguiformer achieved a significant improvement in performance, suggesting that incorporating intrinsic and explanation information can help the model better understand latent patterns that are challenging to learn from the small dataset. Despite being trained from scratch, our model exhibits superior performance compared to the pretrained BIOT and self-supervised Corr-DCRNN model, which implies our model can effectively capture more valuable information through the dRoFE and

Table 3: Ablation studies on dRoFE algorithm and XAI guided transformer.

| Datasets | Metric | XAIguiFormer | w/o dRoFE | w/o XAI guided |
|---|---|---|---|---|
| TUAB | BAC | **0.8205 ± 0.0027** | 0.8018 ± 0.0048 | 0.8070 ± 0.0066 |
| | AUC-PR | 0.8965 ± 0.0079 | 0.8791 ± 0.0046 | **0.8971 ± 0.0047** |
| | AUROC | **0.9000 ± 0.0046** | 0.8846 ± 0.0047 | 0.8997 ± 0.0032 |
| TDBRAIN | BAC | **0.6635 ± 0.0080** | 0.6249 ± 0.0097 | 0.6357 ± 0.0060 |
| | AUC-PR | **0.5961 ± 0.0136** | 0.5135 ± 0.0092 | 0.5347 ± 0.0164 |
| | AUROC | **0.7814 ± 0.0156** | 0.6983 ± 0.0096 | 0.7101 ± 0.0131 |

XAI modules. Furthermore, while the majority of baseline methods are the black box, the XAIgui-iFormer can elucidate its decision-making process, thereby enhancing interpretability alongside its performance.

## 5.4 ABLATION STUDIES

To analyze the effectiveness of our proposed dRoFE and XAI guided transformer, we conducted ablation experiments on both datasets as illustrated in Table 3. In general, it is observed that both algorithms contribute to the overall performance improvement.

**Effectiveness of dRoFE.** To rigorously verify the effectiveness of dRoFE, we replace dRoFE with RoPE for our model instead of simply removing it. This setting ensures a fair comparison, as the transformer without any positional information would inherently lack a sense of position, skewing the results. Table 3 reveals that dRoFE leads to a significant enhancement in performance, achieving +1.87% BAC on the TUAB and +3.86% BAC on the TDBRAIN. This suggests that incorporating frequency and demographic information is more beneficial for the frequency sequence than simple relative position. As a result, dRoFE can effectively capture more comprehensive information and mitigate the influence of individual differences.

**Effectiveness of XAI Guided Transformer.** In this experiment, we ablate the XAI guided self-attention mechanism and exclude the XAI guided item from the entire loss function. As shown in Table 3, the inclusion of XAI guidance results in a BAC increase of +1.35% on the TUAB and +2.78% on the TDBRAIN, demonstrating that XAI does contribute to enhancing the performance of the model by introducing additional knowledge. Interestingly, the TDBRAIN dataset benefits more from XAI guidance. This is likely because the model trained on a small dataset relies more on supplementary information to learn complex representations. In contrast, the model trained on a larger dataset, like TUAB, can learn relatively general knowledge and thus reduce its dependence on the additional knowledge.

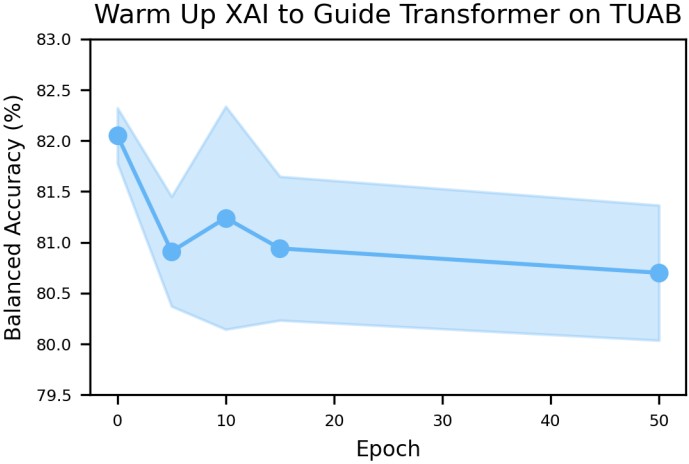

Figure 2: Warmup start XAI to guide transformer, which shows average balanced accuracy and standard deviation across five random seeds on the TUAB. Each point on the line, from left to right, corresponds to a different model configuration where XAI is activated at 0% (normal XAIgui-iFormer), 10%, 20%, 30%, and 100% (without XAI guidance) of the total epochs.

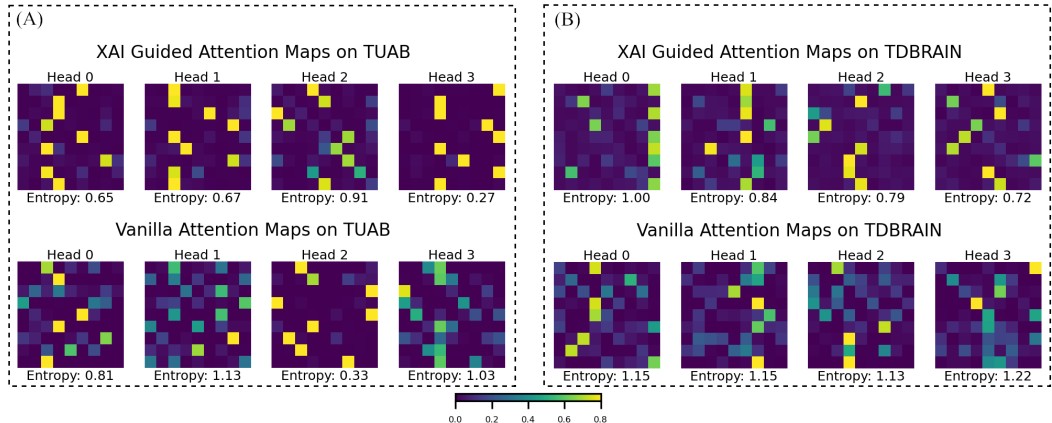

Figure 3: Comparison of attention maps between XAI guided multi-head self-attention and vanilla multi-head self-attention. As shown in the figure, XAI guided attention exhibits a more refined pattern, suggesting a concentration on more crucial dependency relationships and the elimination of distractions from irrelevant tokens. Attention entropy, as detailed in Appendix C, is used as a quantitative measure to assess the concentration of attention maps. Smaller attention entropy indicates that the attention is more concentrated. To facilitate visual comparison, an identical colorbar is used for each attention map. In each attention map, the rows from top to bottom and columns from left to right represent the frequency sequence $\delta, \theta, low\ \alpha, high\ \alpha, low\ \beta, mid\ \beta, high\ \beta, low\ \gamma, \theta/\beta\ ratio$.

**Warmup Start XAI Guidance.** We hypothesize that XAIguiFormer benefits from explanations derived from a relatively good source model. Since the source model would improve during training, we explore a warmup start strategy where XAI guidance is activated after 10%, 20%, and 30% of the total training epochs on the TUAB dataset. As illustrated in Figure 2, while warm-started XAI guidance consistently outperforms the vanilla transformer, it achieves lower performance compared to starting XAI from the beginning of training. This is because activating XAI guidance during training alters the parameter optimization space of the model, as reflected by the observed jitter in the loss function when starting XAI guidance. Despite this, it indirectly verifies the effectiveness of XAI in enhancing model performance.

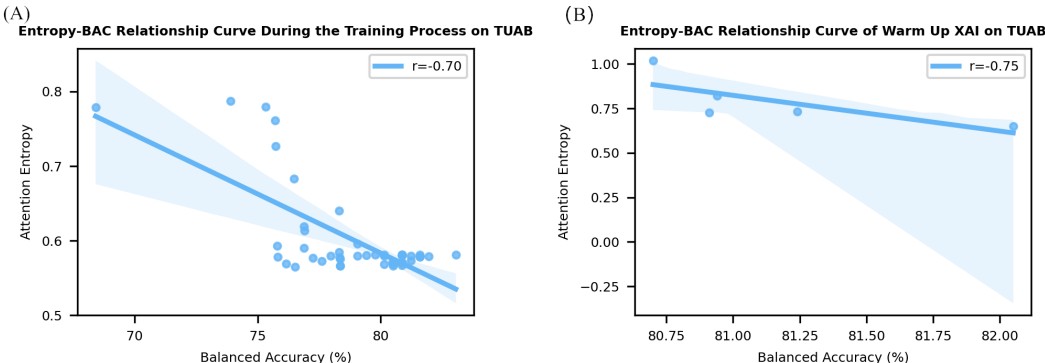

Figure 4: The attention entropy and BAC relationship curve on the TUAB dataset. (A) The attention entropy and BAC relationship under different epochs. The pair of attention entropy and BAC is obtained during the training process. Each point on the scatter plot corresponds to an entropy-BAC pair at a specific training epoch. (B) The attention entropy and BAC relationship under different warmup XAI models. The points from left to right correspond to the activation of XAI at 100% (without XAI), 10%, 30%, 20%, 0% (normal XAIguiFormer) of the total training epoch.

## 5.5 CONCENTRATED ATTENTION PATTERN

Figure 3 visually compares the attention maps generated by the XAI guided and vanilla multi-head self-attention mechanisms. It is evident that XAI guided attention maps focus more on specific dependency relationships, whereas the vanilla attention maps appear more scattered. This improvement is because XAIguiFormer can calculate a more precise attention score based on the refined query and key. Consequently, XAIguiFormer captures core dependency relationships and generates

adaptive values from the perspective of the dynamical system, leading to more effective representations.

To further explore the impact of concentrated attention on the model performance, we conduct experiments to quantitatively evaluate the relationship between attention entropy and BAC under different scenarios. Specifically, we analyzed the entropy-BAC pairs from (1) different training epochs within the same XAIguiFormer model (Figure 4A) and (2) different warm-up XAI models (Figure 4B). These results show that the correaltion coefficients between attention entropy and BAC are -0.7 and -0.75, respectively. This negative correlation suggests that more concentrated attention, characterized by lower entropy, is associated with improved performance.

## 5.6 VISUALIZATION OF INTERPRETABILITY

Figure 5 represents the importance of different frequency bands in our model for the TUAB and TDBRAIN datasets. The frequency band importance can be directly extracted from the explainer used in XAIguiFormer. As depicted, the theta/beta ratio significantly contributes to our model. In psychiatry, this ratio serves as a biomarker for attentional control (Putman et al., 2014), stress assessment (Wen & Aris, 2020) and anxious inhibition (Putman et al., 2010). Consequently, it is able to capture functional aspects of normal and abnormal subject behavior (Schutter & Kenemans, 2022). The low and high alpha bands also play a crucial role in our model, though their functions differ slightly (Debnath et al., 2021). The connectome of high alpha is negatively correlated with elevated mood, whereas the amplitude in low alpha is positively correlated (Petsche et al., 1997). This distinction is why we split the alpha band into subbands. Furthermore, the analysis of narrow EEG bands has proven to be more useful for assessing EEG alterations (Ponomareva et al., 2014). Overall, identifying these frequency bands significantly enhances the interpretability of our model.

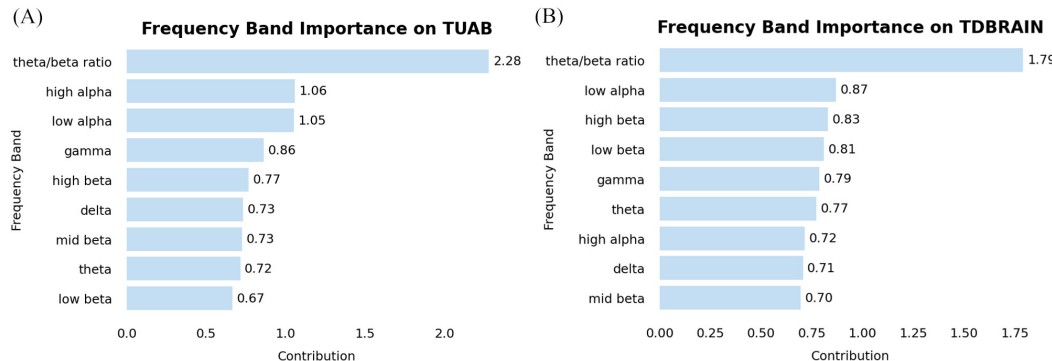

Figure 5: Frequency band importance on the TUAB and TDBRAIN datasets

## 6 CONCLUSION

In this paper, we propose a theoretically inspired architecture, XAI guided transformer (XAIguiiFormer) that not only provides trustworthy explanations but also leverages XAI insights to improve the performance of the transformer. XAIguiFormer takes advantage of multi-band connectomes constructed from EEG signals as input. By treating the single-band graph as an atomic token, the connectome tokenizer preserves the interaction between brain regions. To overcome the model's difficulty in learning the intrinsic information of EEG data, we introduce dRoFE to integrate the intrinsic frequency and demographic information into tokens, thereby mitigating the negative impact of individual differences. Our experiment demonstrates that XAIguiFormer outperforms all baseline methods in the brain disorder classification task. Future efforts can explore self-supervised pretraining of larger models and expand to additional EEG tasks. Finally, we hope this research can draw attention to employing XAI to enhance EEG-based deep learning models, extending its benefits beyond visualization.

ACKNOWLEDGMENTS

The authors gratefully acknowledge the Gauss Centre for Supercomputing e.V. (www.gauss-centre.eu) for funding this project by providing computing time through the John von Neumann Institute for Computing (NIC) on the GCS Supercomputer JUWELS at Jülich Supercomputing Centre (JSC).

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

# A    DEMOGRAPHICS, PREPROCESSING AND CONNECTOME CONSTRUCTION

## A.1    DEMOGRAPHIC DISTRIBUTION

As demographic information plays a crucial role in our proposed dRoFE, it is essential to ensure unbiased dataset splits on demographic distribution. For TUAB dataset, we follow the official train and eval splits. We further split the official train set randomly into our train and val sets, while using the official eval set as our test set. For TDBRAIN, we randomly split the entire dataset into train/val/test sets. Tables 4, 5 and 6 illustrate the demographic distributions (e.g., age, gender, and brain disorder categories) across the train/val/test splits for TUAB and TDBRAIN. Overall, the distributions of age, gender, and brain disorder status are balanced across the splits, minimizing bias and ensuring fair model evaluation.

Table 4: Gender statistics of the TUAB dataset.

| Splits | Abnormal | Normal |
|---|---|---|
| train | 602 (M) vs. 609 (F) | 544 (M) vs. 690(F) |
| val | 65 (M) vs. 70 (F) | 59 (M) vs. 78 (F) |
| test | 63 (M) vs. 63 (F) | 65 (M) vs. 85 (F) |

Table 5: Gender statistics of the TDBRAIN dataset.

| Splits | ADHD | MDD | OCD |
|---|---|---|---|
| train | 89 (M) vs. 42(F) | 117 (M) vs. 110(F) | 21 (M) vs. 17(F) |
| val | 13 (M) vs. 15 (F) | 13 (M) vs. 36 (F) | 6 (M) vs. 2(F) |
| test | 22 (M) vs. 6 (F) | 25 (M) vs. 24 (F) | 5 (M) vs. 3(F) |

Table 6: Age distributions of the TUAB and TDBRAIN datasets.

| Age distributions | TUAB | | | TDBRAIN | | |
|---|---|---|---|---|---|---|
| | train | val | test | train | val | test |
| 0-10 | 7 | 1 | 0 | 34 | 5 | 6 |
| 10-20 | 59 | 3 | 6 | 52 | 9 | 8 |
| 20-30 | 360 | 32 | 34 | 63 | 15 | 18 |
| 30-40 | 321 | 41 | 49 | 69 | 21 | 18 |
| 40-50 | 490 | 45 | 52 | 81 | 16 | 20 |
| 50-60 | 511 | 59 | 51 | 59 | 12 | 11 |
| 60-70 | 381 | 51 | 32 | 27 | 4 | 4 |
| >70 | 316 | 40 | 52 | 11 | 3 | 0 |

## A.2    PREPROCESSING

The raw EEG signals are preprocessed with MNE-Python 1.2.1 (Gramfort et al., 2013), following these steps: Initially, bad channels are automatically detected and removed using pyprep 0.4.2 (Bigdely-Shamlo et al., 2015; Appelhoff et al., 2022). Subsequently, the signals are filtered with a 1-45 Hz band-pass zero-phase Hamming windowed sinc FIR filter (-6dB). The filtered signals are resampled to 250 Hz and segmented into 30-second samples. Next, these samples are subjected to independent component analysis (ICA) using the Infomax algorithm and then artifactual components identified by the pretrained classifier MNE-ICLabel 0.4 (Li et al., 2022a) are rejected. The previously detected bad channels are then repaired using the spherical spline interpolation. Finally, artifact-free samples are recomputed against the average reference.

## A.3    CONNECTOME CONSTRUCTION

The multi-frequency band connectomes are constructed using MNE-Connectivity 0.5.0. First, the 30-second preprocessed signals are further segmented into ten 3-second epochs. For these 3-second

epochs, two complementary methods—coherence and weighted phase lag index (wPLI) are employed to estimate channel-level functional connectivity on the following 9 frequency bands: delta (2-4Hz), theta (4-8Hz), low alpha (8-10Hz), high alpha (10-12Hz), low beta (12-18Hz), mid beta(18-21Hz), high beta (21-30Hz), low gamma (30-45Hz) as well as the theta/beta ratio serving as a reliable biomarker of attention deficit hyperactivity disorder (Angelidis et al., 2016). Next, the functional connectivity from both methods in each frequency band are averaged over the ten epochs. The motivation for employing two complementary construction methods lies in their ability to capture distinct brain activity patterns (Li et al., 2021), thereby providing multi-view information. Finally, we aggregate two types of functional connectivity by using coherence to build node features and wPLI to build edge connections, generating the multi-frequency band connectomes.

## B    BASELINES AND METRICS

To comprehensively compare our method with relevant baseline methods, we consider the following models: (i) FFCL (Li et al., 2022b) fuses multilevel spatial–temporal features based on CNN and LSTM architecture, (ii) SPaRCNet (Jing et al., 2023) is a 1D-CNN based model with dense residual connections, (iii) BIOT (Yang et al., 2024) is a biosignal transformer that learns embeddings from EEG signals of various formats, (iv) S3T (Song et al., 2021) is a EEG transformer to perceive the spatial and temporal features for decoding EEG signals, (v) LaBraM (Jiang et al., 2024) is a pre-trained foundation model that can handle diverse EEG signals with varying channels and lengths, (vi) Corr-DCRNN (Tang et al., 2022) is a self-supervised graph neural network that captures dynamic brain connectivity, (vii) LGGNet (Ding et al., 2024) is a neurologically inspired graph neural network to learn local-global-graph representations from EEG.

To evaluate the performance, we use the following metrics: balanced accuracy (BAC), area under precision-recall curve (AUC-PR) and area under the receiver operating characteristic curve (AU-ROC) for the TUAB and TDBRAIN datasets. We report the average performance and standard deviation across five different random seeds on the TUAB and TDBRAIN datasets. In all scenarios, the best models are trained on the training set, selected from the validation set, and evaluated on the test set. The BAC is employed as the monitor scores to select the best model for the TUAB and TDBRAIN datasets.

## C    ATTENTION ENTROPY

Although Figure 3 clearly shows that XAI guided self-attention is more concentrated, we introduce the attention entropy to quantitatively measure the concentration of attention maps. The attention entropy for each token (row) is defined as

$$E(A_i) = - \sum_{j=1}^{|Freq|} A_{i,j} \log(A_{i,j}) \tag{15}$$

where $A$ represents the attention map. The overall attention entropy for each head is averaged over all token-wise attention entropy. A small attention entropy suggests that the attention tends to be concentrated while a large attention entropy indicates that the attention is more distributed across various tokens.

As illustrated in Figure 3 and the attention entropy, XAI guided self-attention maps are generally more concentrated than vanilla attention maps. This suggests that XAI guided self-attention is better at focusing attention on the most relevant dependency relationships, as the attention score is calculated based on underlying important features. In contrast, the lack of concentration in the vanilla self-attention can lead to the failure to extract the most relevant information.

## D    DATA AUGMENTATION

Mixup (Zhang et al., 2018) is a popular data augmentation technique in computer vision and natural language processing, where random pairs of data samples and their corresponding labels are linearly interpolates to generate synthetic data. However, directly applying Mixup to graph data remains a challenging problem due to the characteristics of graphs: they have irregular node, not well-aligned node order and unique typology in non-Euclidean space. Although the multi-band connectomes

in this study have regular node numbers and aligned node order, it is essential to ensure that the generated connectomes preserve the key topological features of the source connectomes during data augmentation. Therefore, $\mathcal{G}$-Mixup (Han et al., 2022) is employed to augment connectomes, which linearly interpolates the estimated graphons from graphs and then generate synthetic graphs from the mixed graphons. Formally,$\mathcal{G}$-Mixup can be described as

$$\text{Graphon Estimation:} \quad \mathcal{G}_1 \to W_{\mathcal{G}_1}, \mathcal{G}_2 \to W_{\mathcal{G}_2} \tag{16}$$

$$\text{Graphon Mixup:} \quad W_{\mathcal{I}} = \lambda W_{\mathcal{G}_1} + (1 - \lambda)W_{\mathcal{G}_2} \tag{17}$$

$$\text{Graph Generation:} \quad \{I_1, I_2, \cdots, I_n\} \overset{\text{i.i.d}}{\sim} \mathbb{G}(k, W_{\mathcal{I}}) \tag{18}$$

$$\text{Label Mixup:} \quad y_{\mathcal{I}} = \lambda y_{\mathcal{G}_1} + (1 - \lambda)y_{\mathcal{G}_2} \tag{19}$$

where $W_{\mathcal{G}_1}$ and $W_{\mathcal{G}_2}$ are the estimated graphons, $\lambda$ controls the contribution for both estimated graphons, $\mathcal{I} = \{I_1, I_2, \cdots, I_n\}$ is the set of synthetic graphs generated from mixed graphon $W_{\mathcal{I}}$, $\mathbb{G}(k, W_{\mathcal{I}})$ is a random graph with $k$ nodes, and the $y_{\mathcal{G}_1}$ and $y_{\mathcal{G}_2}$ are the ground-truth labels.

Moreover, inspired by $\mathcal{G}$-Mixup and CutMix (Yun et al., 2019), we extend this strategy into $\mathcal{G}$-CutMix. Instead of applying CutMix directly to non-Euclidean graph data, we replace the removed structure in the graphon $W_{\mathcal{G}_1}$ with the corresponding structure from $W_{\mathcal{G}_2}$, ensuring that the augmented connectomes retain critical topological features from the source connectomes. Finally, both $\mathcal{G}$-Mixup and $\mathcal{G}$-CutMix are employed as the data augmentation strategies in the XAIguiFormer.

## E  ROBUSTNESS ANALYSIS OF DIFFERENT EXPLAINERS

To assess the robustness of different XAI algorithms in the XAI guided attention mechanism, we employ two additional popular XAI methods, GradCAM and Integrated Gradients, as explainers. Tables 7 and 8 present the performance achieved by these XAI methods on the TUAB and TD-BRAIN datasets. The results indicate that substituting the original DeepLift does not result in a significant decline in performance. In fact, a slight improvement in the overall performance is observed when DeepLift is replaced with GradCAM. Consequently, XAIguiFormer is able to achieve stable performance without being over reliant on a specific XAI algorithm. In addition, the time complexity of different XAI methods impacts the computational efficiency of XAIguiFormer. Considering both performance and efficiency, GradCAM and DeepLift are recommended as preferred explainers for the XAIguiFormer.

Table 7: Robustness analysis of different explainers on the TUAB dataset.

| Explainers | FLOPs | TUAB | | |
|---|---|---|---|---|
| | | BAC | AUC-PR | AUROC |
| DeepLift | 1.6G | $0.8205 \pm 0.0027$ | $\mathbf{0.8965 \pm 0.0079}$ | $0.9000 \pm 0.0046$ |
| GradCAM | 0.95G | $\mathbf{0.8240 \pm 0.0082}$ | $0.8963 \pm 0.0039$ | $\mathbf{0.9010 \pm 0.0030}$ |
| Integrated Gradients | 35.7G | $0.8210 \pm 0.0047$ | $0.8923 \pm 0.0069$ | $0.8964 \pm 0.0051$ |

Table 8: Robustness analysis of different explainers on the TDBRAIN dataset.

| Explainers | FLOPs | TDBRAIN | | |
|---|---|---|---|---|
| | | BAC | AUC-PR | AUROC |
| DeepLift | 1.6G | $\mathbf{0.6635 \pm 0.0080}$ | $0.5961 \pm 0.0136$ | $0.7814 \pm 0.0156$ |
| GradCAM | 0.95G | $0.6553 \pm 0.0163$ | $\mathbf{0.6149 \pm 0.0155}$ | $\mathbf{0.7996 \pm 0.0143}$ |
| Integrated Gradients | 35.7G | $0.6569 \pm 0.0138$ | $0.5979 \pm 0.0231$ | $0.7874 \pm 0.0194$ |

Furthermore, we evaluate the robustness of the calculated frequency band importance by changing the XAI algorithms. Figures 6 and 7 illustrate the frequency band importance generated by Grad-CAM and Integrated Gradients, respectively. The rankings of the theta/beta ratio, high and low $\alpha$ on TUAB remain consistent across different XAI methods. Similarly, on the TDBRAIN dataset, the rankings of the theta/beta ratio, low $\alpha$ and high $\beta$ are largely consistent. These results demonstrate that the ranking of the most important frequency bands remains stable, indicating strong robustness across different XAI algorithms.

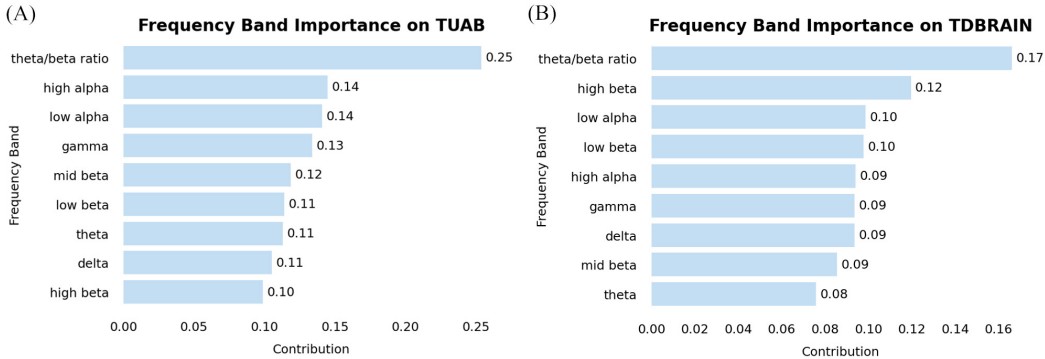

Figure 6: Frequency band importance generated by GradCAM on the TUAB and TDBRAIN datasets

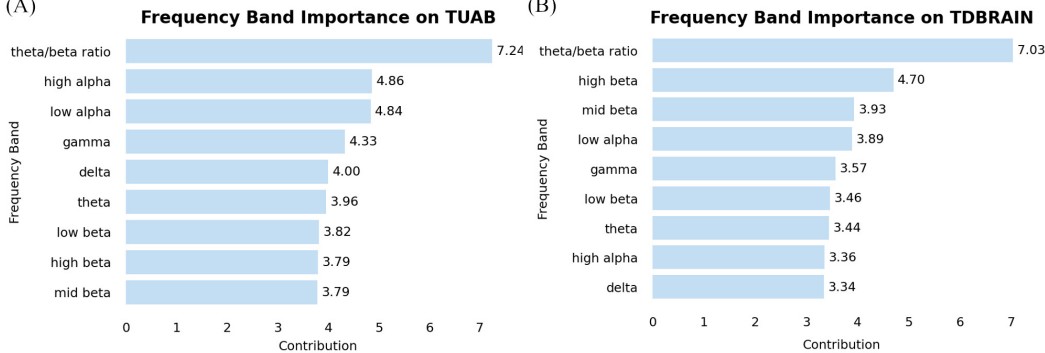

Figure 7: Frequency band importance generated by Integrated Gradients on the TUAB and TD-BRAIN datasets

# F ABLATION STUDIES ON THE CONTROLLING PARAMETER $\alpha$

To investigate the impact of the controlling parameter $\alpha$ in the XAI guided loss function, we conduct a series of experiments with different settings ranging from 0.1 to 0.9 in increments of 0.1. Figure 8 presents the mean and standard deviation of the BAC across five different random seeds on the TUAB and TDBRAIN datasets, where GradCAM is employed as the explainer within XAIgui-iFormer. The results indicate that the optimal $\alpha$ value is 0.8 for TUAB and 0.9 for TDBRAIN. However, when $alpha$ reaches 0.9 on TUAB, the BAC begins to decline. Therefore, we conclude that $alpha$ = 0.7 or 0.8 serves as a generally effective setting for both datasets. In addition, these experimental results indicate that when $\alpha$ is lower than 0.3, the performance declines significantly. This is likely because, with a low $\alpha$, the explainer's guidance becomes insufficient to meaningfully influence the learning process, leading to degraded performance.

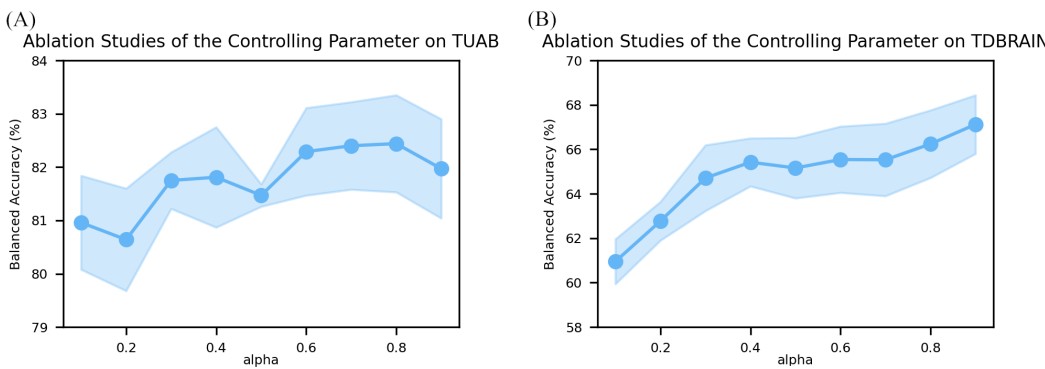

Figure 8: Ablation studies of the controlling parameter $\alpha$ on the TUAB and TDBRAIN datasets

## G    HYPERPARAMETER CONFIGURATIONS

Table 9: Hyperparameter configurations of XAIguiFormer on the TUAB and TDBRAIN datasets.

| **Hyperparameters** | | **TUAB** | **TDBRAIN** |
|---|---|---|---|
| Connectome tokenizer | Input dimensions | 19 | 26 |
| | Hidden dimensions | 128 | 128 |
| | Layer number | 4 | 4 |
| XAIguiFormer | Head number | 4 | 4 |
| | MLP ratio | 4 | 4 |
| | Layer scale init | 1e-3 | 1e-3 |
| | Layer number | 12 | 12 |
| Training settings | Optimizer | AdamW | AdamW |
| | Adam $\beta$ | (0.9,0.99) | (0.9,0.99) |
| | Peak learning rate | 4e-5 | 5e-5 |
| | Warmup learning rate | 1e-7 | 1e-7 |
| | Learning rate scheduler | Cosine | Cosine |
| | Batch size | 512 | 64 |
| | Weight decay | 1e-4 | 1e-5 |
| | Warmup epochs | 5 | 5 |
| | Training epochs | 50 | 100 |
| | Label smoothing | 0.1 | 0.1 |
| | XAI guided loss weight | 0.7 | 0.7 |
| Data augmentation | $\mathcal{G}$-Mixup probability | 1.0 | 1.0 |
| | $\mathcal{G}$-CutMix probability | 0.5 | 0.5 |

## H    LIMITATIONS AND OUTLOOK

First, although we have evaluated XAIguiFormer on two large-scale clinical datasets (TUAB and TDBRAIN), they may not fully capture the diversity of real-world clinical scenarios. Our work represents an initial step toward exploring the positive impact of explanation-guided learning on enhancing performance in the EEG field. Second, it should be noted that XAIguiFormer currently provides explanations at the frequency band level, rather than at the level of functional connectivity. This limitation arises because XAIguiFormer focuses on token importance (query, key, and value vectors) to calculate refined attention values, which restricts explanations to the token level. As a result, the explanations are confined to the token (frequency band) level and do not offer detailed insights into the functional connectivity patterns. In the future, we plan to explore additional clinical datasets to further verify the effectiveness of our model. Additionally, we aim to explore pretraining strategies for XAIguiFormer to further improve its performance.

