# OpenReview forum: "XAIguiFormer: explainable artificial intelligence guided transformer for brain disorder identification"
_ICLR.cc/2025/Conference — ICLR 2025 Poster_

### Official Review · Reviewer_vgdo · 2024-10-29

**Soundness:** 3
**Presentation:** 2
**Contribution:** 3
**Rating:** 6
**Confidence:** 4

**Summary:**

Authors proposed XAIguiFormer, the first framework to employ XAI for enhancing transformer performance in neuroimaging data. The explanability and accuracy are both improved as shown in experiments, while the improvement of accuracy might be not caused by the higher explanability.

**Strengths:**

1. Clear motivations.

2. Good paper writing.

3. Idea of explanability guiding self-attention is novel.

**Weaknesses:**

1. Figures are too far to the corresponding descriptive text, e.g., fig.2 in page 6 is described in page 9. I recommend moving Figure 2 closer to its description or adding a forward reference earlier in the text.

2. The attention map by the proposed XAI has lower entropy, which is called an improvement in line 463. It confused readers to an accuracy improvement according to the proposed XAI attention map. Although authors were selling the story under such motivation, the relationship between entropy and accuracy are missed. Can you provide entropy-accuracy curves or other quantitative evidence demonstrating how the lower entropy attention maps directly contribute to improved model performance? I think this the key results to support the proposal in this paper, I will raise the score if these results are fit with the expectation.

3. The proposed method leads to a new attention map that is more determined and hence easier to be explained. Even there is an qualitative assessment given neuroscience literatures, more details are needed on how the results in fig.4 were computed. Can you include a quantitative evaluation of how well the attention maps align with established neuroscience knowledge?

**Questions:**

See weakness.

---

> ### Author Response · Authors · 2024-11-23
>
> We sincerely appreciate your constructive suggestion to thoroughly evaluate the effectiveness of our XAI-guided attention mechanism. We have conducted additional experiments to quantitatively assess the relationship between attention entropy and model performance.
>
> > **[W1]** Figures are too far to the corresponding descriptive text, e.g., fig.2 in page 6 is described in page 9. I recommend moving Figure 2 closer to its description or adding a forward reference earlier in the text.
>
> R1: We apologize for the inconvenience caused by the placement of Figure 2. We have relocated Figure 2 to page 7, positioning it closer to the corresponding descriptive text to improve readability. Additionally, we have added a forward reference earlier in the text to ensure that readers can easily locate the figure if needed.
>
> > **[W2]** The attention map by the proposed XAI has lower entropy, which is called an improvement in line 463. It confused readers to an accuracy improvement according to the proposed XAI attention map. Although authors were selling the story under such motivation, the relationship between entropy and accuracy are missed. Can you provide entropy-accuracy curves or other quantitative evidence demonstrating how the lower entropy attention maps directly contribute to improved model performance? I think this the key results to support the proposal in this paper, I will raise the score if these results are fit with the expectation.
>
> R2: Thank you for highlighting this important concern regarding the relationship between attention entropy and model performance. To address this issue, we conducted additional experiments to quantitatively evaluate the relationship between attention entropy and Balanced Accuracy (BAC) under various scenarios. We analyzed the entropy-BAC pairs from (1) different warm-up XAI models and (2) different training epochs within the same XAIguiFormer model. The correlation coefficients between attention entropy and BAC in these scenarios were found to be -0.7 and -0.75, respectively. This negative correlation indicates that lower attention entropy (more concentrated attention) is consistently associated with improved performance. The detailed entropy-BAC curves are included in Appendix B of the revised manuscript.
>
> On the other hand, a recently popular work [1], Differential Transformer, supports the hypothesis that sparse/concentrated (lower-entropy) attention patterns can improve performance. Differential Transformer argues that conventional attention mechanisms in Transformers tend to overallocate attention to irrelevant contexts. By introducing a differential approach to amplify attention toward relevant contexts while suppressing noise, the model encourages sparse and concentrated attention patterns, resulting in improved performance. Similarly, the XAI-guided attention in our method compels the Transformer to concentrate on sparse and relevant patterns through the XAI approach, thereby reducing attention entropy and filtering out irrelevant information.
>
>
> Reference:
>
> [1] Ye, Tianzhu, et al. Differential transformer. Submit to ICLR 2025.

---

> > ### Author Response · Authors · 2024-11-23
> >
> > > **[W3]** The proposed method leads to a new attention map that is more determined and hence easier to be explained. Even there is an qualitative assessment given neuroscience literatures, more details are needed on how the results in fig.4 were computed. Can you include a quantitative evaluation of how well the attention maps align with established neuroscience knowledge?
> >
> > R3: Thank you for this insightful suggestion. Indeed, the concentrated attention map is easier to explain. However, the significance of the frequency bands in Figure 4 is not derived directly from the attention map but rather from the explainer integrated within XAIguiFormer. The explainer calculates the importance scores of the input tokens, where each token corresponds to a single frequency band connectome. These token-level importance scores are then aggregated to produce a corresponding frequency band score, which reflects its contribution to the model's decision-making process.
> >
> > In this context, a quantitative evaluation of the alignment between the importance of the frequency bands and neuroscience knowledge necessitates a formal metric. Unfortunately, existing tools such as Neurosynth, which calculate correlations between explanations and neuroscience knowledge, are limited to brain regions in fMRI studies and do not extend to frequency bands in EEG. Developing a similar framework for EEG frequency bands would require the curation of a comprehensive database of neuroscience findings, which is beyond the scope of this paper and could serve as the foundation for a separate research project. In addition to the qualitative assessment, the consistency of frequency band importance derived from different XAI methods presented in Appendix E offers indirect evidence for the validity and robustness of the explanations.
> > If the reviewer has any specific suggestions or alternative ideas for conducting a quantitative evaluation, we would be pleased to explore and implement them.

---

> ### Comment · Reviewer_vgdo · 2024-11-23
>
> **[W2].a**: Given results of the additional experiments shown in Fig.5, the proposed method is empirically proved. But I have to suggest the authors arrange Fig. 5 into the main text since there is enough white space in your manuscript, where detailed methodology steps can be kept in Appendix. Furthermore, it looks like a logistic regression (e.g., `seaborn.regplot`) can fit better than the linear regression authors currently used. I recommend that you improve the presentation accordingly.
>
> **[W2].b**: Listing the evidence shown in previous works to support your motivation is effective in improving the soundness of your proposed methods. Instead of an ICLR submission, there are published peer-reviewed works you can review to do this, e.g., [1] has solid evidence to support sparse attention can improve accuracy in both empirical and theoretical aspects. I recommend you include it in the Introduction to enhance the motivation and proposal of this paper.
>
> [1] NeuroPath: A Neural Pathway Transformer for Joining the Dots of Human Connectomes. NeurIPS, 2024, https://openreview.net/forum?id=AvBuK8Ezrg
>
> Authors have revised their work with additional empirical evidence to support the proposal in this paper via [W2].a. They also have shown an aspiration of side evidence via [W2].b. My concern in [W3] cannot be resolved by quantitative evidence at this point. Conclusively, I'd like to raise my score if authors can arrange their responses into the manuscript.

---

> > ### Author Response · Authors · 2024-11-24
> >
> > We sincerely appreciate your prompt and detailed comments, which have been invaluable in improving the quality of our manuscript.
> >
> > > **[W2].a**: Given results of the additional experiments shown in Fig.5, the proposed method is empirically proved. But I have to suggest the authors arrange Fig. 5 into the main text since there is enough white space in your manuscript, where detailed methodology steps can be kept in Appendix. Furthermore, it looks like a logistic regression (e.g., seaborn.regplot) can fit better than the linear regression authors currently used. I recommend that you improve the presentation accordingly.
> >
> > R1: Thank you for your valuable feedback. Following your suggestion, we have relocated Figure 5 and its descriptive text from the Appendix into the main text, ensuring better accessibility and improving the paper's readability. Additionally, we have adjusted the placement of the original Figure 2, now located on page 9, so that it appears alongside its corresponding descriptive text for a more intuitive presentation.
> >
> > Regarding the entropy-BAC curve, we agree that a logistic regression fit may visually capture the trend better. However, logistic regression is specifically designed for binary classification tasks and does not provide an R-value for assessing the relationship between the continuous variables, attention entropy and BAC. For this reason, Figure 5 in the revised manuscript has been updated with a new fitted curve with confidence intervals using the linear regression of seaborn.
> >
> > To address your suggestion comprehensively, we have included an additional entropy-BAC curve fitted with logistic regression (without R-value) in the supplementary material for comparison.
> >
> > > **[W2].b**: Listing the evidence shown in previous works to support your motivation is effective in improving the soundness of your proposed methods. Instead of an ICLR submission, there are published peer-reviewed works you can review to do this, e.g., [1] has solid evidence to support sparse attention can improve accuracy in both empirical and theoretical aspects. I recommend you include it in the Introduction to enhance the motivation and proposal of this paper.
> >
> > R2: Thank you for highlighting this important aspect and referencing the relevant work. We have included a discussion of the referenced work in the Introduction of the revised manuscript to further strengthen the motivation and rationality of our proposed method.

---

### Official Review · Reviewer_LoKT · 2024-10-30

**Soundness:** 3
**Presentation:** 2
**Contribution:** 3
**Rating:** 8
**Confidence:** 4

**Summary:**

The paper introduces XAIguiFormer, a transformer model that uses explainable AI to both interpret and improve brain disorder detection from EEG data. The model features novel tokenization and encoding methods to preserve brain network patterns, achieving better performance than existing approaches while providing insights into which brain wave frequencies are most important for diagnosis.

**Strengths:**

The paper uses XAI to actively improve model performance rather than just for interpretation.

The connectome tokennization approach and demographic-aware frequency encoding are also novel.

There are comprehensive experiments and ablation studies.

**Weaknesses:**

Authors should discuss further on why they chose to model EEG as connectome rather than time series, given some state-of-the-art EEG foundation models with time series as input, such as [1], especially it focuses on frequency domain as well. Experimental comparisons with these EEG models are also missed in the paper.

It seems like only temporal frequency is considered in positional encoding, how about spatial frequency for different brain regions? Some work like [2] developed spatial connectome based positioning, which should at least be discussed in the paper.

Limited dataset scope: Only tested on two datasets (TUAB and TDBRAIN) which may not fully represent real-world clinical diversity.

Discussion of training time, computational efficiency compared to the baselines would be appreciated.

The model relies heavily on DeepLift's accuracy without exploring alternative explanation methods ot validating explanation quality.

[1] Large Brain Model for Learning Generic Representations with Tremendous EEG Data in BCI. ICLR 2024.

[2] Brain-JEPA: Brain Dynamics Foundation Model with Gradient Positioning and Spatiotemporal Masking. NeurIPS 2024.

**Questions:**

The proposed method assumes explanations from an imperfectly trained model can improve performance, would that propagate early training errors?

---

> ### Author Response · Authors · 2024-11-23
>
> We appreciate the opportunity to clarify and expand some experiments to improve our model. Please find below our point-by-point responses to your comments.
>
> > **[W1]** Authors should discuss further on why they chose to model EEG as connectome rather than time series, given some state-of-the-art EEG foundation models with time series as input, such as [2], especially it focuses on frequency domain as well. Experimental comparisons with these EEG models are also missed in the paper.
>
> R1: Thank you for your insightful comment. There are two primary reasons for employing the connectome as input. First, compared to time series data, functional connectivity offers a superior advantage in modeling the interactions between channels or brain regions, as the brain is a complex communication and information processing system [1]. Additionally, we constructed the connectome by aggregating two distinct types of functional connectivity that provide complementary information, thereby providing multi-view information. Second, when developing XAIguiFormer, we carefully considered the additional computational burden introduced by the XAI method. Due to the nature of EEG multi-channel data, patching multi-variable time series into tokens results in a relatively long sequence length, which increases the computational demands of the XAI module. By utilizing the connectome as input and implementing a specially designed connectome tokenizer, we were able to reduce the sequence length, thereby enhancing computational efficiency while preserving essential information.
>
> In our comparative experiments, we evaluated our approach against the S3T model and the pretrained BIOT model, both of which utilize EEG time series data. Following the reviewer’s suggestion, we also included the time series model LaBraM [2] as a baseline and conducted the comparison experiment.
>
> | Methods| Model Size|FLOPs  ||**TUAB**||
> |--|--|--|--|--|--|
> ||||BAC|AUC-PR|AUROC|
> | LaBraM-Base|5.8M|2.7G|0.8140 ± 0.0019| 0.8965 ± 0.0016 |**0.9022 ± 0.0009**|
> | XAIguiFormer(Ours)|3.5M|1.6G|**0.8205 ± 0.0027**| 0.8965 ± 0.0079 |0.9000 ± 0.0046|
>
> | Methods| Model Size|FLOPs  ||**TDBRAIN**||
> |--|--|--|--|--|--|
> ||||BAC|AUC-PR|AUROC|
> | LaBraM-Base|5.8M|2.7G|0.6456 ± 0.0089| 0.5438 ± 0.0058 |0.7147 ± 0.0145|
> | XAIguiFormer(Ours)|3.5M|1.6G|**0.6635 ± 0.0080**| **0.5961 ± 0.0136** |**0.7814 ± 0.0156**|
>
> > **[W2]** It seems like only temporal frequency is considered in positional encoding, how about spatial frequency for different brain regions? Some work like [3] developed spatial connectome based positioning, which should at least be discussed in the paper.
>
> R2: Thank you for highlighting this important aspect and referencing the relevant work. The Brain Gradient Positioning method proposed by Brain-JEPA incorporates both temporal and spatial information into tokens by introducing functional connectivity gradients. This approach captures the functional relationships among brain regions and integrates them with temporal information from fMRI time series segments to encode positional information. The input to the transformer in Brain-JEPA consists of time series segments from multiple brain regions, thereby creating a “two-dimensional” structure. One dimension represents the spatial and functional relationships among regions, while the other dimension is the temporal information of the time series segments. In this context, both spatial and temporal information are essential intrinsic characteristics of fMRI data.
>
> In contrast, the input tokens of the transformer in XAIguiFormer are generated from the connectome constructed in the frequency domain. These tokens **encapsulate** the spatial relationships among the channels through the connectome structure and connectome tokenizer. Unlike multi-channel or multi-region time-series tokens, the frequency band connectome tokens in our method do not inherently contain spatial relationships. Consequently, spatial information is less critical in our framework compared to frequency and demographic information, which are explicitly prioritized to enhance model performance. Additionally, we include and discuss the Brain-JEPA as related work in our paper.
>
> Reference:
>
> [1] Seguin C, Sporns O, Zalesky A. Brain network communication: concepts, models and applications[J]. Nature reviews neuroscience, 2023, 24(9): 557-574.
>
> [2] Jiang W, Zhao L, Lu B. Large Brain Model for Learning Generic Representations with Tremendous EEG Data in BCI. ICLR 2024.
>
> [3] Dong Z, Ruilin L, Wu Y, et al. Brain-JEPA: Brain Dynamics Foundation Model with Gradient Positioning and Spatiotemporal Masking. NeurIPS 2024.

---

> > ### Author Response · Authors · 2024-11-23
> >
> > > **[W3]** Limited dataset scope: Only tested on two datasets (TUAB and TDBRAIN) which may not fully represent real-world clinical diversity.
> >
> > R2: Thank you for raising this important concern regarding the scope of the dataset. TUAB and TDBRAIN are two large-scale datasets used in clinical EEG research. They encompass a variety of recording conditions and patient demographics, providing a meaningful foundation for evaluating our model to a certain extent. However, we acknowledge that these datasets may not fully capture the diversity of real-world clinical scenarios. Due to time constraints, particularly in conducting numerous supplementary experiments, we find it challenging to incorporate additional datasets for evaluating our model at this time. We have included this limitation in the revised manuscript and highlighted our intention to validate the model on additional datasets with broader clinical diversity in future work.
> >
> > > **[W4]** Discussion of training time, computational efficiency compared to the baselines would be appreciated.
> >
> > R4: Thank you for your suggestion regarding computational efficiency. As discussed in R1, the use of the connectome structure and the specially designed connectome tokenizer mitigates the additional computational burden introduced by the XAI method. To compare computational efficiency, we employed floating point operations (FLOPs) rather than training time. FLOPs provide a hardware-independent metric that remains unaffected by variations in GPU performance or system configurations, offering a more consistent and fair comparison. Based on FLOPs, XAIguiFormer demonstrates superior computational efficiency compared to several transformer-based baselines, such as LaBraM and BIOT. However, it is worth noting that the specific XAI method utilized within XAIguiFormer can influence its computational efficiency (see more details in R5).
> > |Methods|FFCL|SPaRCNet|BIOT|S3T|LaBraM-Base|Corr-DCRNN|LGGNet|XAIguiFormer(Ours)|
> > |--|--|--|--|--|--|--|--|--|
> > |**FLOPs**|0.83G|0.26G|1.9G|0.22G|2.7G|0.21G|0.64G|1.6 G|
> >
> > > **[W5]** The model relies heavily on DeepLift's accuracy without exploring alternative explanation methods or validating explanation quality.
> >
> > R5: Thank you for emphasizing the importance of evaluating the reliance on DeepLift and exploring alternative explanation methods. To assess DeepLift’s dependence and evaluate the robustness of different XAI algorithms within the XAI guided attention mechanism, we employ two additional popular XAI methods, GradCAM and Integrated Gradients, as explainers. Our results indicate that substituting the original DeepLift with these methods does not result in a significant decline in performance. Interestingly, we observe a slight improvement in the overall performance when DeepLift is replaced with GradCAM. Consequently, XAIguiFormer is able to achieve stable performance without being highly dependent on a specific XAI algorithm.
> >
> > Furthermore, we also assess the robustness of the calculated importance of frequency bands. Figures 6 and 7 in Appendix E illustrate the frequency band importance generated by GradCAM and Integrated Gradients, respectively. The rankings of the theta/beta ratio, as well as high and low $\alpha$ on TUAB remain consistent across different explainers. Similarly, on the TDBRAIN dataset, the rankings of the theta/beta ratio, low $\alpha$ and high $\beta$ are largely consistent,  where low $\alpha$ ranks fourth in importance identified by Integrated Gradients. These results demonstrate that the ranking of the most important frequency bands remains stable, indicating strong robustness across different XAI algorithms.
> >
> > | Explainers| FLOPs  ||**TUAB**||
> > |--|--|--|--|--|
> > |||BAC|AUC-PR|AUROC|
> > | DeepLift|1.6G|0.8205 ± 0.0027| **0.8965 ± 0.0079** |0.9000 ± 0.0046|
> > | GradCAM|0.95G| **0.8240 ± 0.0082** |0.8963 ± 0.0039|**0.9010 ± 0.0030**|
> > | Integrated Gradients|35.7G |0.8210 ± 0.0047|0.8923 ± 0.0069|0.8964 ± 0.0051|
> >
> > | Explainers| FLOPs  ||**TDBRAIN**||
> > |--|--|--|--|--|
> > |||BAC|AUC-PR|AUROC|
> > | DeepLift|1.6G| **0.6635 ± 0.0080**|0.5961 ± 0.0136|0.7814 ± 0.0156|
> > | GradCAM|0.95G|0.6553 ± 0.0163|**0.6149 ± 0.0155**|**0.7996 ± 0.0143**|
> > | Integrated Gradients|35.7G|0.6569 ± 0.0138|0.5979 ± 0.0231 |0.7874 ± 0.0194|

---

> > > ### Author Response · Authors · 2024-11-23
> > >
> > > > **[Q1]** The proposed method assumes explanations from an imperfectly trained model can improve performance, would that propagate early training errors?
> > >
> > > R6: This is an important question. In our original manuscript, we hypothesized that XAIguiFormer benefits from explanations derived from a relatively good source model, as we believe that poor explanations could propagate early training errors. To investigate this, we conducted an experiment in which the XAI module was warmed up later in the training process, rather than activated from the start. Contrary to our expectations, the results did not demonstrate a significant improvement when the activation of the XAI module was delayed. One possible explanation is that activating the XAI module at a later training stage may alter the optimization space, making it more challenging to train the model compared to activating the XAI module from the beginning of the training process.

---

> > > > ### Comment · Reviewer_LoKT · 2024-11-23
> > > >
> > > > Thank authors for comprehensive responses that thoroughly address the key points raised during review.
> > > >
> > > > It is good to see the additional comparison with EEG foundation model and further discussion on input tokens. Experiments on addtional explainers, and analysis on the robustness of the calculated importance of frequency bands, significantly strengthen the paper's arguments.
> > > >
> > > > These responses have satisfactorily addressed my previous concerns, and I have increased my score to 8 accordingly.

---

### Official Review · Reviewer_D4N1 · 2024-11-02

**Soundness:** 3
**Presentation:** 3
**Contribution:** 3
**Rating:** 6
**Confidence:** 3

**Summary:**

The paper introduces XAIguiFormer, a transformer model guided by explainable AI (XAI) techniques to enhance model performance. The authors integrate both frequency and demographic information into the model’s tokenization process, demonstrating that these features are essential for the observed performance improvements. XAIguiFormer achieves superior results compared to baseline models

**Strengths:**

1. The authors clearly define the specific issue they aim to address and effectively outline their solution approach. The introduction is clear and acknowledges the relevant literature and highlighting the limitations of current methods. The authors also provide a thorough comparison with a range of other methods and evaluate their approach on two public datasets. Another strength is that they intend to share their code upon acceptance, promoting transparency and reproducibility. The ablation studies also highlights the importance of both proposed suggestion (positional embedding and models guided by interpretability)

**Weaknesses:**

2. Section 5.1 provides a brief description of the datasets; however, given the importance of demographic information to the proposed method, the authors should expand on this aspect in that section. Additionally, it would be helpful to know if demographics were considered when creating the train/test/validation split. How was the data split? Are the different diseases balanced on both datasets? Furthermore, what is the distribution of patients with brain disorders versus healthy individuals across splits?

**Questions:**

3. Line 369: The authors mention that the BAC, AUC-PR are the average performance and standard deviation across five different random seeds on the TUAB and TDBRAIN dataset. I am assuming that the models were re-trained 5 times using different train/val splits while test data was kept constant, could the authors clarify if this is case? I am guessing that the reason why the models were re-trained 5 times is due to computational limitations, could the authors elaborate on the computational costs of their method compared to the other baseline methods?
4. In the text line 335 the authors mention: “XAIguiFormer is not sensitive to \alpha as long as it is larger than 0.3”. Could the authors report those results and discuss why they believe that there is a lower threshold but not an upper threshold?
5. Figure 4 illustrates the importance of difference frequency bands, could the author further clarity how the frequency information can be extracted from XAIguiFormer?
6. Could the authors include some discussion on the limitations of the presented method?

---

> ### Author Response · Authors · 2024-11-23
>
> We appreciate your suggestion and believe that this additional information enhances the clarity and comprehensiveness of XAIguiFormer.
>
> > **[W1]** Section 5.1 provides a brief description of the datasets; however, given the importance of demographic information to the proposed method, the authors should expand on this aspect in that section. Additionally, it would be helpful to know if demographics were considered when creating the train/test/validation split. How was the data split? Are the different diseases balanced on both datasets? Furthermore, what is the distribution of patients with brain disorders versus healthy individuals across splits?
>
> R1: Thank you for pointing out this important consideration, as demographic information plays a crucial role in our proposed dRoFE. For the TUAB dataset, we followed the official train and eval splits. We further split the official training set randomly into our train and val sets, while using the official eval set as our test set. For TDBRAIN, we randomly split the entire dataset into train/val/test sets. As shown in these tables, the distributions of age, gender, and brain disorder status are balanced across the splits, thereby minimizing bias and ensuring fair model evaluation.
>
> |Splits|**TUAB** ||**TDBRAIN**|||
> |--|--|--|--|--|--|
> |    |Abnormal|Normal|ADHD|MDD|OCD|
> |train|602(M) vs. 609(F)|544(M) vs. 690(F)|89(M) vs. 42(F)|117(M) vs. 110(F)| 21(M) vs. 17(F)|
> |val|65(M) vs. 70(F)|59(M) vs. 78(F)|13(M) vs. 15(F)|13(M) vs. 36(F)|6(M) vs.2(F)|
> |test|63(M) vs. 63(F)|65(M) vs. 85(F)|22(M) vs. 6(F)|25(M) vs. 24(F)|5(M) vs. 3(F)|
>
>
> | Age distributions ||**TUAB** |||**TDBRAIN**||
> |--|--|--|--|--|--|--|
> |    | train | val | test            | train | val | test |
> | 0-10  |7 |1| 0|34| 5| 6|
> | 10-20|59| 3| 6|52| 9| 8|
> | 20-30|360|32|34|63| 15| 18|
> | 30-40|321|41|49|69| 21| 18|
> | 40-50|490|45|52|81| 16| 20 |
> | 50-60|511|59|51|59| 12| 11 |
> | 60-70|381|51|32|27|4 | 4 |
> | >70|316|40 |52|11| 3| 0|
>
> > **[Q1]** Line 369: The authors mention that the BAC, AUC-PR are the average performance and standard deviation across five different random seeds on the TUAB and TDBRAIN datasets. I am assuming that the models were re-trained 5 times using different train/val splits while test data was kept constant, could the authors clarify if this is case? I am guessing that the reason why the models were re-trained 5 times is due to computational limitations, could the authors elaborate on the computational costs of their method compared to the other baseline methods?
>
> R2: Unlike k-fold cross-validation, we employed a hold-out strategy to split the datasets into train, val, and test sets, where all splits remain constant across experiments. The reason for training the model five times is to assess the stability of the model under different random seeds. This stability reflects the model's robustness to variations in random weight initialization and data batch ordering, both of which are influenced by the random seed.
>
> When developing XAIguiFormer, we carefully considered the additional computational burden caused by the introduction of the XAI method. Therefore, we employed the connectome as input and designed a specialized connectome tokenizer to reduce the sequence length, thereby improving computational efficiency while preserving essential information. XAIguiFormer demonstrates better efficiency (lower FLOPs) than transformer-based baselines, such as LaBraM and BIOT, despite the inclusion of the additional computation cost from the XAI algorithm.
>
> |Methods|FFCL|SPaRCNet|BIOT|S3T|LaBraM-Base|Corr-DCRNN|LGGNet|XAIguiFormer(Ours)|
> |--|--|--|--|--|--|--|--|--|
> |**FLOPs**|0.83G|0.26G|1.9G|0.22G|2.7G|0.21G|0.64G|1.6 G|
>
> > **[Q2]** In the text line 335 the authors mention: “XAIguiFormer is not sensitive to \alpha as long as it is larger than 0.3”. Could the authors report those results and discuss why they believe that there is a lower threshold but not an upper threshold?
>
> R3: Thank you for raising this point. We have included a detailed relationship curve between $\alpha$ and BAC in Appendix F, where $\alpha$ varies from 0.1 to 0.9 in increments of 0.1. The experimental results indicate that when $\alpha$ is lower than 0.3, the performance declines significantly. This is likely because, with a low $\alpha$, the explainer’s guidance becomes insufficient to meaningfully influence the learning process, leading to degraded performance. On the other hand, our results show no upper threshold led to a significant performance drop. This could be because larger $\alpha$ values allow the explainer's guidance to dominate the attention mechanism without overwhelming the model's ability to learn effectively.

---

> > ### Author Response · Authors · 2024-11-23
> >
> > > **[Q3]** Figure 4 illustrates the importance of difference frequency bands, could the author further clarity how the frequency information can be extracted from XAIguiFormer?
> >
> > R4: Thank you for your question. In the XAIguiFormer, we employ post-hoc explanation methods such as DeepLift, GradCAM, and Integrated Gradients as explainers to generate explanations of each layer for the XAI-guided attention mechanism. After training the model, we use the explainer to compute the importance of the input tokens, where each token corresponds to a single frequency band connectome. The importance scores for each token are then aggregated to produce a corresponding frequency band score, which reflects its contribution to the model's decision-making process.
> >
> > > **[Q4]** Could the authors include some discussion on the limitations of the presented method?
> >
> > R5: Thank you for your suggestion. We have supplemented the Limitations and Outlook section in the revised manuscript. First, while we have evaluated XAIguiFormer on two large-scale clinical datasets (TUAB and TDBRAIN), these datasets may not fully capture the diversity of real-world clinical scenarios. Second, XAIguiFormer currently provides explanations at the frequency band level, rather than at the level of functional connectivity. This limitation arises because XAIguiFormer focuses on token importance (query, key, and value vectors) to calculate refined attention values, which restricts explanations to the token level. As a result, the explanations are confined to the token (frequency band) level and do not offer detailed insights into the functional connectivity patterns.

---

### Official Review · Reviewer_A5ot · 2024-11-04

**Soundness:** 3
**Presentation:** 3
**Contribution:** 3
**Rating:** 6
**Confidence:** 4

**Summary:**

This work proposed a dynamical-system-inspired architecture, XAI guided transformer (XAIguiFormer), where XAI not only provides explanations but also contributes to enhancing the transformer by refining the originally coarse information in the self-attention mechanism to capture more relevant dependency relationships.

**Strengths:**

1. This method proposed a fusion strategy to explicitly inject the frequency and demographic information into tokens, improving the model’s understanding of the frequency and mitigating the negative effects of individual differences.
2. This method proposed to use XAI to directly enhance transformer performance rather than focusing only on analyzing the visual interpretability.

**Weaknesses:**

1. Robustness is unclear. How about the performance after changing to other explainers in the module in addition to using DeepLift? How about the robustness?
2. How to calculate the frequency band importance? Is the result from the explainer inside the proposed model? If so, can it get the same conclusion after changing the explainer?
3. In Fig 3, why does the accuracy of warm-started XAI keep decreasing after training? The accuracy at the beginning is the best one, so how does it prove the effectiveness of the proposed method? It becomes worse after training the proposed one.

**Questions:**

see above. The questions are about the description of warmup start XAI guidance and the robustness of the explainer module in the proposed method.

---

> ### Author Response · Authors · 2024-11-23
>
> We sincerely appreciate your thorough review and valuable comments, which have significantly contributed to enhancing the robustness of XAIguiFormer and clarifying our description of warmup XAI. Please find below our point-by-point responses to your comments.
>
> > **[W1]** Robustness is unclear. How about the performance after changing to other explainers in the module in addition to using DeepLift? How about the robustness?
>
> R1: Thank you for your insightful comment regarding the robustness of our method with respect to different explainers. To address this, we conducted additional experiments by replacing the original DeepLift explainer with two widely used alternatives: GradCAM and Integrated Gradients. Our results show that the replacement of DeepLift does not result in a significant performance degradation. Interestingly, we observed a slight improvement in the overall performance when GradCAM was utilized instead of DeepLift. These findings highlight that XAIguiFormer is robust and performs stably across various explainers, indicating that its effectiveness is not overly dependent on a specific XAI algorithm.
>
> | Explainers| FLOPs  ||**TUAB**||
> |--|--|--|--|--|
> |||BAC|AUC-PR|AUROC|
> | DeepLift|1.6G|0.8205 ± 0.0027| **0.8965 ± 0.0079** |0.9000 ± 0.0046|
> | GradCAM|0.95G| **0.8240 ± 0.0082** |0.8963 ± 0.0039|**0.9010 ± 0.0030**|
> | Integrated Gradients|35.7G |0.8210 ± 0.0047|0.8923 ± 0.0069|0.8964 ± 0.0051|
>
>
> | Explainers| FLOPs  ||**TDBRAIN**||
> |--|--|--|--|--|
> |||BAC|AUC-PR|AUROC|
> | DeepLift|1.6G| **0.6635 ± 0.0080**|0.5961 ± 0.0136|0.7814 ± 0.0156|
> | GradCAM|0.95G|0.6553 ± 0.0163|**0.6149 ± 0.0155**|**0.7996 ± 0.0143**|
> | Integrated Gradients|35.7G|0.6569 ± 0.0138|0.5979 ± 0.0231 |0.7874 ± 0.0194|
>
> > **[W2]** How to calculate the frequency band importance? Is the result from the explainer inside the proposed model? If so, can it get the same conclusion after changing the explainer?
>
> R2: The importance of the frequency band is derived from the explainer within the XAIguiFormer. To assess the robustness of the calculated frequency band importance, we replaced the original explainer (DeepLift) with GradCAM and Integrated Gradients in XAIguiFormer and extracted the frequency band importance for comparison. Figures 6 and 7 in Appendix E illustrate the frequency band importance generated by GradCAM and Integrated Gradients, respectively. The rankings of the theta/beta ratio, high and low $\alpha$ on TUAB remain consistent across different explainers. Similarly, on the TDBRAIN dataset, the rankings of the theta/beta ratio, low $\alpha$ and high $\beta$ are largely consistent, where low $\alpha$ ranks fourth in importance identified by Integrated Gradients. These results demonstrate that the ranking of the most important frequency bands remains stable, indicating strong robustness across different XAI algorithms.
>
> > **[W3]** In Fig 3, why does the accuracy of warm-started XAI keep decreasing after training? The accuracy at the beginning is the best one, so how does it prove the effectiveness of the proposed method? It becomes worse after training the proposed one.
>
> R3: We apologize for any confusion and are pleased to provide clarification. Figure 3 is not a training curve for a single model utilizing warm-started XAI. Instead, it represents a line chart demonstrating the relationship between different warm-started XAI models and their balanced accuracy (BAC) after training. Each point on the line, from left to right, corresponds to a different model configuration, where XAI is activated at 0% (normal XAIguiFormer), 10%, 20%, 30%, and 100% (without XAI) of the total epochs during the course of training process. The y-axis represents the final BAC for each model upon completion of training. Since Figure 3 presents independent final results for each configuration, it does not imply that the accuracy of a single model decreases over the course of training. The results depicted in Figure 3 demonstrate that models with warm-started XAI guidance consistently outperform the vanilla transformer, thereby validating the effectiveness of XAI in enhancing model performance.
>
> > **[Q1]** see above. The questions are about the description of warmup start XAI guidance and the robustness of the explainer module in the proposed method.
>
> R4: We hope our responses above have addressed reviewer’s concerns on the robustness of the explainer module and clarified the concepts of warmup start XAI guidance.

---

> > ### Author Response · Authors · 2024-11-27
> >
> > Dear Reviewer A5ot,
> >
> > We hope this message finds you well. Thank you for your valuable comments and suggestions, which have greatly contributed to improving the robustness of XAIguiFormer. We have carefully addressed the concerns you raised and posted detailed responses to each of them.
> >
> > We understand that this may be a particularly busy time, and we sincerely appreciate any time you can spare to review our responses and provide further feedback. If you have additional questions or suggestions, we would be happy to discuss them further. Your insights and feedback are very important to us, and we want to ensure we have addressed all your comments thoroughly and effectively.
> >
> > Thank you again for the time and effort you dedicated to reviewing this work.
> >
> >
> > Best regards,
> >
> > The Authors

---

### Author Response · Authors · 2024-12-02

We would like to express our sincere gratitude to all the reviewers for their insightful feedback and thoughtful evaluation of our work. We appreciate their recognition of our contributions, including **the novel demographic-aware rotary frequency encoding** (@A5ot, D4N1, LoKT), **the innovative application of XAI to improve model performance rather than solely for interpretation** (@A5ot, D4N1, LoKT, vgdo), and **the thorough experiments and ablation studies** (@D4N1, LoKT) that **effectively underscore the significance of the proposed methods** (@D4N1).

During the rebuttal, we carefully addressed the reviewers' primary concerns and included additional experiments and clarifications. Below, we provide a brief summary of the key improvements:

**[Robustness of different XAI algorithms @A5ot, LoKT]** We conducted additional experiments by replacing the original DeepLift explainer within the XAI guided attention mechanism with two widely used alternatives: GradCAM and Integrated Gradients. We found that the performance did not significantly decline. In fact, replacing DeepLift with GradCAM resulted in a slight improvement in overall performance. Furthermore, we evaluated the stability of the calculated frequency band importance and confirmed that the most significant frequency bands remained consistent across different explainers. This robustness has been **acknowledged and endorsed** by Reviewer LoKT, who actively participated in the discussion.

**[Computational efficiency @D4N1, LoKT]** To mitigate the computational burden introduced by XAI methods, we utilized the connectome as input and developed a specialized connectome tokenizer to reduce sequence length while preserving essential information. This approach enables XAIguiFormer to achieve better efficiency (lower FLOPs) than transformer-based baselines (e.g., LaBraM, BIOT).

**[Calculation of frequency band importance @D4N1, vgdo]** In XAIguiFormer, post-hoc explanation methods such as DeepLift, GradCAM, and Integrated Gradients are employed to generate layer-wise explanations for the XAI guided attention mechanism. After training, the explainer computes the importance of input tokens, where each token corresponds to a specific frequency band connectome. These importance scores are aggregated to produce a frequency band score that reflects its contribution to the model’s decision-making process.

**[Contribution of concentrated attention to the performance @vgdo]** We conducted additional experiments to quantitatively evaluate the relationship between attention entropy and Balanced Accuracy (BAC) across various scenarios. Specifically, we analyzed the entropy-BAC pairs from (1) different warm-up XAI models and (2) different training epochs within the same XAIguiFormer model. The correlation coefficients of -0.7 and -0.75, respectively, indicate a strong negative correlation, confirming that lower entropy (indicating concentrated attention) improves performance.

Overall, our manuscript has been improved, and we deeply appreciate the reviewers for their time and effort. We welcome any additional comments and feedback on our work.

---

### Meta-Review · Area_Chair_b1D9 · 2024-12-21

**Metareview:**

The authors propose a guided transformer architecture that uses explainable AI methods to improve and interpret brain disorder detection using EEG data. The paper provides good motivation for their approach and comprehensive expderiments with ablation studies to verify their method. The use of explainability guiding self-attention was a novel contribution.

Four reviewers assessed the paper and all recommended acceptance. During the discussion, the authors were able to address concerns well, with the result that three of four reviewers increased their score. It is recommended that, space permitting, the content provided in the responses to the reviewers be incorporated in the final draft.

**Additional Comments On Reviewer Discussion:**

The reviewers engaged with the authors during the rebuttal and were satisfied with the response, to the point that three of them increased their scores. All reviewers consider this paper above threshold and their improved sentiment about the contributions of the method to XAI indicate it should be accepted with information from the response worked into the final draft.

---

### Decision · Program_Chairs · 2025-01-22

Accept (Poster)